# Pancreatic Neuroendocrine Tumors: Signaling Pathways and Epigenetic Regulation

**DOI:** 10.3390/ijms25021331

**Published:** 2024-01-22

**Authors:** Zena Saleh, Matthew C. Moccia, Zachary Ladd, Upasana Joneja, Yahui Li, Francis Spitz, Young Ki Hong, Tao Gao

**Affiliations:** 1Department of Surgery, Cooper University Health Care, Camden, NJ 08103, USA; saleh-zena@cooperhealth.edu (Z.S.); laddza45@rowan.edu (Z.L.);; 2Department of Pathology, Cooper University Health Care, Camden, NJ 08103, USA; 3Camden Cancer Research Center, Camden, NJ 08103, USA

**Keywords:** pancreatic neuroendocrine tumors, signaling pathways, epigenetic regulation

## Abstract

Pancreatic neuroendocrine tumors (PNETs) are characterized by dysregulated signaling pathways that are crucial for tumor formation and progression. The efficacy of traditional therapies is limited, particularly in the treatment of PNETs at an advanced stage. Epigenetic alterations profoundly impact the activity of signaling pathways in cancer development, offering potential opportunities for drug development. There is currently a lack of extensive research on epigenetic regulation in PNETs. To fill this gap, we first summarize major signaling events that are involved in PNET development. Then, we discuss the epigenetic regulation of these signaling pathways in the context of both PNETs and commonly occurring—and therefore more extensively studied—malignancies. Finally, we will offer a perspective on the future research direction of the PNET epigenome and its potential applications in patient care.

## 1. Introduction

Pancreatic neuroendocrine tumors (PNETs) are a rare and heterogeneous group of neoplasms arising from pancreatic islet cells. They account for approximately 2% of all pancreatic malignancies and are characterized by their slow-growing nature and potential for metastasis [1]. Current treatment options for PNETs are limited, and the prognosis for patients with advanced disease remains poor, underscoring the urgent need for innovative therapeutic strategies [2]. PNETs were originally classified into three distinct categories: (1) G1, which are well-differentiated PNETs and characterized by mitoses <2/10 high-power field (HPF) and Ki67 index <3%; (2) G2, which are well-differentiated PNETs with mitoses ranging from 2 to 20/10 HPF or Ki67 index between 3% and 20%; (3) G3, which are poorly differentiated and also called pancreatic neuroendocrine carcinomas (PNECs) with mitoses exceeding 20/10 HPF or Ki67 index surpassing 20%. Over the years, advancements in clinical practice and research, particularly in next-generation sequencing and analysis, have led to the recognition that the G3 group comprises at least two distinct subtypes—the well-differentiated PNET G3 and poorly differentiated PNECs—which are genetically different from each other [3]. In 2019, the W.H.O. revised the grading classification of pancreatic neuroendocrine neoplasms (PNENs) (5th edition) with a pivotal change being the division of the original G3 group into PNET G3 and PNECs. This new classification aims to clarify the distinction between these clinically and molecularly separate entities. The genetic features of well-differentiated nonfunctional PNETs and poorly differentiated PNECs significantly differ, contributing to variations in tumor formation, growth, and progression. Poorly differentiated PNECs frequently exhibit mutations in the TP53 and RB1 genes, causing enhanced invasion and metastasis capabilities. Conversely, patients with highly differentiated nonfunctional PNETs more commonly show genetic alterations, including ATRX, DAXX, and MEN1 [4,5,6,7,8]. It is crucial to emphasize that while PNETs and PNECs typically display distinctions in genomic mutations, there are sporadic cases where both diseases share mutations in specific genes like p53, CDKN2A, RB1, and KRAS, as revealed by genetic studies of PNET and PNEC samples, which further underscore the complexity of the cancer origin [9,10]. Therefore, in this context, we have integrated pathways that are common to both PNETs and PNECs into this manuscript. This inclusion aims to provide the audience with a more comprehensive understanding of molecular signaling in PNETs.

The pathogenesis of PNETs is intricate and characterized by complex interactions among numerous signaling pathways. Each pathway plays a role in different facets of tumor development, encompassing cell proliferation, survival, migration, and angiogenesis. Together, these pathways constitute a complex network that, when disrupted, can result in uncontrolled tumor growth and metastasis [11]. Germline and somatic whole-genome sequencing provided a comprehensive analysis of PNET-related genetic variations [4]. Most PNETs occur sporadically, with only around 10% of the cases associated with germline mutations. Germline mutations such as multiple endocrine neoplasia type 1 (MEN1), von Hippel–Lindau syndrome (VHL), neurofibromatosis type 1 (NF1), and the occasionally tuberous sclerosis complex (TSC) are the most identified PNET-associated mutations [12,13]. For sporadically occurring PNETs, MEN1, DAXX (death domain associated protein), ATRX (α-thalassemia/mental retardation syndrome X-linked), and genes related to the mammalian target of rapamycin (mTOR) pathway harbor commonly identified somatic alterations [13].

While somatic and germline mutations remain of great significance for diagnosis and therapeutic treatment, emerging evidence highlights the pivotal role of epigenetic modifications in shaping the intricate landscape of PNET-related signaling [14]. Epigenetic modifications encompass a range of reversible alterations that modulate gene expression patterns without affecting the underlying DNA sequence, such as DNA methylation, histone modifications, and non-coding RNAs (ncRNAs), which coordinate together to govern cellular processes crucial for normal development and homeostasis [15]. In the context of PNETs, epigenetic dysregulation is starting to gain recognition as a significant player in the initiation and evolution of the disease [1,16,17]. Epigenetic alterations can lead to aberrant gene expression patterns, contributing to uncontrolled cell growth, evasion of cell death, and increased metastatic potential—the hallmark characteristics of cancer [18]. This is particularly relevant in the case of PNET because PNETs have a low tumor mutational burden and are considered an epigenetic disorder due to the fact that multiple high-frequency mutations, including MEN1, DAXX, and ATRX, are all found involved in epigenetic regulation [14].

Several recent reviews have addressed the complicated signaling network in PNET development [1,11,17,19,20]. However, to the best of our knowledge, there is a deficiency in comprehensive studies that systematically review the epigenetic status of PNET-related signaling pathways—considering both PNET and other frequently encountered cancers simultaneously. Given that epigenetic mechanisms are common across various cancers [21,22], conducting a thorough examination of the epigenetic events within signaling pathways in the broader context of cancer research is certain to offer valuable insights. This approach will undoubtedly guide the identification of future directions for the relatively understudied field of epigenetic research in PNETs.

Because of our strong interest in enhancing the care of patients diagnosed with PNETs, we will concentrate on the epigenetic mechanisms that hold the highest clinical relevance. Presently, there are a total of eight FDA-approved anti-tumor epigenetic drugs primarily targeting DNA methylation and histone modifications [23]. Beyond their therapeutic applications, epigenetic markers, particularly in the realm of non-coding RNAs such as microRNAs (miRNAs), play a crucial role in cancer diagnosis [24]. This review will specifically delve into the three aforementioned epigenetic regulatory mechanisms: DNA methylation, histone modifications, and non-coding RNAs. This is by no means all-encompassing, as we recognized that we would not discuss other epigenetic modifications, including RNA methylation, histone ubiquitylation, phosphorylation, SUMOylation, ADP ribosylation, citrullination, and biotinylation at specific amino acid residues [25].

## 2. Major Signaling Pathways in PNETs

Various molecular alterations have been identified as correlated with the development of PNETs. These events manifest through diverse mechanisms, encompassing both genetic and epigenetic modifications, resulting in a complex regulatory network. Genetic variations of PNETs include both familial and predominantly sporadic mutations [26]. Each of the hereditary syndromes, namely MEN1, VHL, NF1, and TSC, is characterized by distinct sets of signaling pathways and molecules [26]. Moreover, numerous signaling pathways exhibit sporadic mutations in PNET samples [26]. It is worth noting that familial and sporadic mutations are not exclusive of each other in PNETs. One such example is MEN1 signaling. MEN1 mutations play a pivotal role in the initiation and progression of PNETs, as over 40% of sporadic PNETs and all MEN1 patients exhibit somatic mutations in the MEN1 gene [27,28]. More interestingly, the status of MEN1 mutations and menin protein expression do not always correlate well with each other. One study, which included the mutational analysis and immunohistochemistry results (IHC) of 169 PNET patients, showed that 80% of sporadic cases showed a loss of menin nuclear localization, while only 25% of the patients carried a mutation in the MEN1 gene itself [29]. This study clearly suggests that other regulatory mechanisms, besides genetic mutations, are involved in the altered level of signaling pathways in PNET pathology. The complicated regulatory mechanisms in PNET pathology have been nicely reviewed previously [1,11,17,19,20]. Here, we adopt a different approach by discussing the PNET-related signaling events based on the major epigenetic regulatory mechanisms that they fall into, including DNA methylation, histone modifications, and ncRNAs (Figure 1). Dynamic alterations in DNA methylation, histone modifications, and ncRNA activity are integral aspects of the pathogenesis of other extensively studied cancers. However, the contributions of these epigenetic modifications to PNET pathology have yet to be fully explored.

## 3. Epigenetic Regulation of PNET-Related Signaling Pathways

### 3.1. DNA Methylation

Methylation status has long been recognized as one of the primary mechanisms by which epigenetic modification is carried out within cells to regulate gene expression. DNA methyl transferases (DNMTs) cause DNA methylation by adding a methyl group to the cytosine in CpG DNA sequences (CpG islands) of promoter or enhancer. In cancer cells, dysregulated CpG methylation leads to promoter hypermethylation and downregulated gene expression, resulting in gene silencing of tumor suppressor genes. In addition, DNA demethylating enzymes might cause genome-wide hypomethylation, causing DNA instability. A variety of tumor suppressors, as well as proto-oncogenes, have been shown to be modified via methylation in tumorigenesis [31].

#### 3.1.1. MEN1

MEN1 is one of the most extensively studied genes in PNETs. Germline mutations in this gene result in Multiple Endocrine Neoplasia Syndrome Type 1, causing different types of cancer, including PNETs, pituitary adenoma, and parathyroid hyperplasia. The genetic mutation responsible for MEN1 spans 9.8 kb of chromosome 11q13. The protein product of MEN1, termed menin, is a tumor suppressor protein ubiquitously expressed in the cell nucleus [32]. Considering menin’s crucial role as an epigenetic regulator, it is unsurprising that the analysis of DNA methylation in MEN1-associated PNETs revealed the promoter hypermethylation of numerous potential tumor suppressor genes [33]. Epigenetic modifications of MEN1 are also identified in non-endocrine tumors. Notably, in advanced breast cancer, the overexpression of the MEN1 gene is associated with promoter hypomethylation, which differs from the scenario observed in PNETs [34].

#### 3.1.2. mTOR-TSC

The mechanistic target of rapamycin (mTOR) is a kinase that, in humans, is encoded by the mTOR gene. It functions as a serine/threonine protein kinase that regulates a variety of cellular functions, including cell growth and proliferation, motility, survival, protein synthesis, autophagy, gene transcription, and thus carcinogenesis. mTOR forms two protein complexes, which include mTORC1, susceptible to rapamycin, and mTORC2, not directly hindered by this drug [35]. PTEN dephosphorylates phosphatidylinositol-3,4,5 trisphosphate (PIP3) to negatively regulate the PI3K-AKT signaling pathway, which acts upstream of the mTOR signaling cascade. The dysregulation of PI3/AKT signaling causes the upregulation of protein synthesis, cell migration, and tumor-induced angiogenesis [36]. Mutations of PTEN are responsible for many cancers including brain, breast, prostate, endometrial carcinoma, head and neck, melanoma, and PNETs [37]. Activation of mTOR signaling can affect serine one-carbon metabolism enough to modulate DNA methylation and promote tumorigenesis [38]. Aberrant hypermethylation of the promoter of tumor suppressor genes, followed by their silencing, is important for aberrant mTOR-PI3K pathway activation in gastric cancer (GC) [39]. Tumor suppressor TRPM4, a calcium-activated nonselective cation channel, inhibits the PI3K/Akt/mTOR signaling pathway to impede tumor migration and invasion. This gene has been shown to be methylated at its promoter with silenced or reduced expression in colorectal cancer (CRC), though this effect can be reversed with the calpain inhibitor calpeptin, providing possible therapeutic strategies for aggressive disease [40]. ZDHHC22, a well-known member of the palmitoyltransferase family, regulates mTOR stability and activation of the AKT signaling pathway. Hypermethylation of the promoter CpG island of the ZDHHC22 gene is associated with poorer prognosis in breast cancer. Overexpression of ZDHHC22 inhibits breast cancer cell growth both in vitro and in vivo [41]. The major signaling molecules of the mTOR pathway, including PTEN, PI3K, AKT, c-Myc, and TSC, have previously been well-documented in PNET pathology [42,43,44,45,46,47,48]. TSC2 forms a complex with TSC1, which negatively regulates mTOR. Dissolution of the TSC2/TSC1 complex allows activation of mTOR [49]. Mutations in either TSC1 or TSC2 lead to the onset of tuberous sclerosis complex, characterized by the formation of hamartomas, also called benign tumors, in the skin, brain, lungs, heart, and kidneys [50]. Patients with TSC are also at greater risk for developing malignancies, such as renal cell carcinoma, breast cancer, thyroid cancer, and PNETs [51]. The promoter of the PTEN gene has been shown to be hypermethylated, while the methylation status of the promoter of TSC2 remains unaltered in PNET samples [52].

#### 3.1.3. Hypoxia-Induced Factor 1α (HIF1α)-VHL

Hypoxia-inducible factor1α (HIF1α) is a pro-tumorigenic factor that has been found to lead to poor prognosis, metastasis, and recurrence in multiple cancer subtypes, including breast, renal cell, gastric, and ovarian cancer. Its involvement in B-cell lymphoma, chronic lymphocytic leukemia (CLL), colon cancer, small-cell lung cancer, and PNETs have also been established [53]. HIF1α is regulated by hypoxia. Under normoxic conditions, HIF1α interacts with the von Hippel–Lindau (VHL) protein (pVHL), resulting in its polyubiquitination and proteasomal degradation [54]. During cellular hypoxia, HIF1α translocates to the nucleus where it becomes stabilized, activating genes involved in cellular proliferation, angiogenesis, glucose homeostasis, and metastasis [55]. In patients with von Hippel–Lindau syndrome, the tumor suppressor gene is mutated and pVHL is absent, limiting HIF1α degradation. Patients with VHL are at increased risk for developing hemangioblastomas, retinal angiomas, renal cell carcinomas, pheochromocytoma, and pancreatic lesions, including PNETs [56]. The methylation status of the HIF binding region within the hypoxia response element correlates with increased expression of epidermal growth factor receptor (EGFR) under hypoxic conditions in breast cancer. These changes can be reversed by treatment with DNA methyltransferase inhibitors such as azacytidine or decitabine, suggesting that patients with hypoxic breast tumors and hypomethylated EGFR status may benefit from EGFR inhibitors [57]. HIF1α-driven transcriptional response in hypoxia in pediatric neuroblastoma is subject to epigenetic control via DNA methylation status of gene regulatory regions. Hypoxia exposure induces global DNA hypermethylation in neuroblastoma cells, and HIF1A itself might control DNA methylation [58]. In VHL disease, the loss of VHL leads to the activation of its binding partner HIF1α, causing multi-organ tumorigenesis including PNETs [59,60]. The VHL gene promoter has been shown to be hypermethylated in a subset of familial PNET samples [61].

#### 3.1.4. RAS-MAPK-NF1

The RAS-MAPK pathway mediates cellular responses to growth signals and is often dysregulated in cancer. The mitogen-activated protein kinase (MAPK) pathway involves extracellular signal-regulated kinases 1 and 2 (ERK1/2) that mediate multiple cellular functions, including proliferation, growth, and senescence [62]. The rat sarcoma (RAS) gene is an oncogene that is part of a large family of GTPases and acts as a control switch for the ERK1/2 pathway. In order to transmit signals to downstream effector proteins, RAS must properly bind to cellular membranes, which requires posttranslational modifications such as DNA methylation and histone acetylation [63]. Global DNA hypermethylation correlated with the RAS and MAPK oncogenic pathways (among others) is significantly associated with high-grade tumors, platinum resistance, and poor prognosis in high-grade serous ovarian carcinoma. Treatment with demethylating agents shows significant growth retardation in ovarian cancer cells through differential inductions such as cell apoptosis or G2/M cell cycle arrest [64]. The status of 5-methylcytosine (m5C) regulators is a robust predictor of prognosis and therapy response in pancreatic ductal adenocarcinoma (PDAC) [65]. Differentially expressed genes, both the hypermethylated–low expression and hypomethylation–high expression, in the Ras/MAPK pathways are seen in neurofibromatosis type 2 vestibular schwannomas [66]. Genomes of acute myeloid leukemia (AML) cells resistant to the BCL-2–selective inhibitor venetoclax (VEN) show extensive differential methylation with the activation of the RAS/MAPK pathway, leading to increased stability and higher levels of MCL-1 protein. VEN sensitivity can then be at least partially restored by the silencing or pharmacologic inhibition of MCL-1 [67]. RASSF1A is a negative regulator of the RAS-MAPK signaling pathway and can mediate DNA repair through the mechanism of nucleotide excision repair [68]. RASSF1A might be the most frequently inactivated tumor suppressor identified in human cancers and has been found to be inactivated in more than 40 types of human malignancies [69]. The expression of RASSF1A is lower in bladder and prostate cancer patients due to promoter hypermethylation. If this is reversed and RASSF1 is overexpressed, enhanced cytotoxicity to chemotherapeutic drugs is observed [70]. Urinary methylation assays can even be used to help predict prostate tumor stage and grade, and related tests utilizing RASSF1A methylation in various contexts have been investigated in endometrial, breast, and CRCs [71,72,73,74]. Methylation of the RASSF1 gene promoter is also significantly higher in lung tumor tissues and is associated with lymph node metastasis and clinical stage [75,76]. RASSF1 promoter methylation, under the impact of the methyltransferase genes DNMT1 and MGMT, can also be a papillary thyroid cancer genetic marker [77]. The promoter of the RASSF1 gene was found to be hypermethylated in 75% of NF-PNETs where its inactivation results in the degradation of beta-catenin [78]. Inhibition of RAS-MAPK attenuates PNET cell growth [79]. Methylation has been associated with RASSF1A downregulation and, interestingly, the upregulation of RASSF1C in PNET samples [80]. Neurofibromatosis type 1 (NF1) encodes a RAS GTPase activating protein called Neurofibromin, which has been found to be mutated in both sporadic cancers and familial syndromes. Mutations of NF1 result in excessive RAS signaling. NF1-associated sporadic cancers include glioblastoma, neuroblastoma, AML, lung cancer, ovarian cancer, and breast cancer [81]. In the NF1 familial syndrome, patients with neurofibromatosis will present with benign cutaneous neurofibromas and are predisposed to breast, gastric, and pancreatic neuroendocrine tumors [17,82]. Despite the significant importance of NF1 in PNET regulation, there is currently no reported information concerning the DNA methylation of NF1 in the progression of PNETs.

#### 3.1.5. ATRX/DAXX

ATRX and DAXX form a histone chaperone complex that deposits histone variant H3.3 into specific genomic regions [83]. Mutations in ATRX constitute the most prevalent genetic abnormalities in gliomas, and interestingly, ATRX alterations are associated with favorable outcomes. CRISPR/Cas9 knockout ATRX glioblastoma cells demonstrated compromised H3K9 trimethylation, which led to decreased DNA repair by the tumor cells. In addition, increased response to DNA damage-inducing agent temozolomide was confirmed in these cells, suggesting that these mutations might serve as a prognostic maker in predicting chemosensitivity [84]. In high-grade meningiomas, there is a significant increase in ATRX expression at both gene and protein levels, particularly in the nuclear compartment, and an increased DAXX protein expression level. In these tumors, ATRX/DAXX is suggested to have a role in telomere maintenance due to the low variability of telomere length [85]. In histopathologic evaluation of human pituitary adenomas, ATRX or DAXX protein loss in the absence of genetic mutations was observed, suggesting the presence of additional silencing mechanisms such as promoter methylation [86]. In PNET samples, mutations in ATRX and DAXX have been identified [42,43,87]. In addition, the DAXX gene has been shown to be hypermethylated at the promoter region in PNET samples, which, along with DAXX genetic mutations, might contribute to PNET progression [88].

#### 3.1.6. CDKN2A-RB1

Abnormal DNA methylation is found in the early stages of kidney cancers. Genistein, a potent antioxidant and demethylating agent in soybean-enriched foods, induces cell apoptosis and inhibits the cell proliferation of multiple types of kidney cancer cells. It achieves this through the expression of CDKN2A via decreased CDKN2A methylation [89]. CDKN2A promoter methylation correlates with a poor prognosis and progression-free survival in both ovarian and CRCs [90,91]. Peroxisome proliferator-activated receptor alpha (PPARα), the molecular target of fibrates commonly used to treat dyslipidemia and diabetes, inhibits DNMT1 activity and abolishes methylation-mediated CDKN2A repression in CRC both in vitro and in vivo, indicating that it may be an applicable agent for epigenetic therapy of colon cancer patients [92]. In addition, there is a strong and significant correlation between CDKN2A gene methylation and the risk of cervical cancer as well as pre-cancerous lesions [93]. Furthermore, the level of CDKN2A gene methylation in human GC samples is increased compared to noncancerous tissues. Variations in methylation patterns and different loci are associated with changes in overall survival [94]. Inducing methylation in GC cells increased their sensitivity to palbociclib, an anti-CDK4/6 chemical for cancer treatment, both in vitro and in vivo [95]. As part of the CDKN2A-CDK4/6-RB1 axis, retinoblastoma tumor suppressor protein1 (RB1) plays critical roles in tumor suppression. RB1 inhibits cellular proliferation by repressing the transcription of genes essential for cell cycle progression, serving as a negative regulator of the cell cycle. More specifically, RB1 inhibits G1 to S transition through the repression of E2F target genes, which are involved in DNA synthesis and cell cycle progression [96]. Aberrant DNA methylation at the RB1 gene promoter has been reported in multiple types of tumors [97]. Members of the CDKN2A-CDK4/6-RB1 pathway, including CDKN2A, P14^ARF^, and RB1 have shown promoter hypermethylation in PNET samples [52,98,99,100], suggesting that the DNA methylation-mediated downregulation of CDKN2A-RB1 signaling might contribute to PNET development.

#### 3.1.7. p53

p53, also known as tumor protein TP53, is a regulatory protein that is often mutated in human cancers. The DNA hypermethylation of TP53 has been found associated with the pathogenesis of cervical cancer in Northeastern Indian patients [101]. Resveratrol, a chemical mostly found in red grapes, alters the methylation of histones in the TP53 promoter in human breast cancer cells and upregulates the expression of SET domain-containing lysine methyltransferase 7/9 (SET7/9) in colorectal cells to positively regulates p53 through its mono-methylation function [102,103]. Luteolin, a dietary flavone molecule, modulates various signaling pathways involved in carcinogenesis and has apoptotic effects mediated by the DNA demethylation of the nuclear factor erythroid 2-related factor 2 (NRF2) promoter and the interaction of Nrf2 and p53 in human colon cancer cells [104]. DNMT3B has been shown to be elevated in nasopharyngeal carcinoma tissues and predicts a poor prognosis. Ionizing radiation can induce DNMT3B, which might contribute to radio-resistance because DNMT3B gene silencing restores p53 via DNA demethylation, leading to cell cycle arrest and apoptosis both in vitro and in vivo [105]. PBX/Knotted Homeobox 2 (PKNOX2) functions as a candidate tumor suppressor, which exerts its tumor-suppressive effect by promoting the upregulation of p53. PKNOX2 mRNA expression is largely silenced in both GC cell lines and primary GC via promoter methylation, which is associated with poor outcomes [106]. The genetic profiling of BON-1 and QGP-1 PNET cell lines showed homozygous TP53 mutations, suggesting that loss-of-function of p53 contributes to PNET pathology [107]. This is consistent with the observation that advanced metastatic PNET patients often display mutations in the TP53 and RB1 locus [87]. In addition, p53 and its downstream effector PHLDA3 have been found to be hypermethylated at their gene promoter regions, respectively [52,108], suggesting that p53 signaling regulates PNET development at both genetic and epigenetic levels.

#### 3.1.8. Notch Signaling

The Notch signaling pathway is a highly conserved cell signaling system present in most animals whose proteins span the cell membrane. In the Notch signaling pathway, the binding of ligand and notch receptor binding re-leases the Notch Intracellular Domain (NICD), which translocates to the nucleus to activate gene expression [109]. Notch signaling is responsible for differentiation and tissue homeostasis, and its dysregulation has been attributed to the development of multiple cancers, including leukemias such as T-cell acute lymphoblastic leukemia (T-ALL) and CLL [110]. In CRC, the upregulation of oncogenic histone cluster 2 H2B family member F (HIST2H2BF) enhances malignancy aggressiveness in humans and increases liver metastasis in mice through the activation of Notch signaling. The reactivation of this cluster is found related to promoter CpG hypomethylation [111]. The high expression of SETD1A, a histone methyltransferase that specifically methylates H3K4, acts as a key oncogene in ovarian cancer and is associated with a poor prognosis. Accordingly, the overexpression of SETD1A augments cell proliferation, migration, and invasion in vitro via initiation of Notch signaling, and the downregulation of SETD1A reduces tumorigenesis in vivo [112]. Among North Indian patients with cervical cancer, the methylation rate of Notch1 and Notch3 promoters is notably elevated compared to healthy tissues, accompanied by a downregulation in protein expression. In addition, Notch1 promoter methylation increases with age, severity of the disease, and HPV infection in patients with cervical cancer [113]. The low expression of the methyltransferase DNMT3A, a key regulator of DNA methylation, is associated with more aggressive types of CLL. Dnmt3a knockout mice consistently developed CLL and showed a general upregulation of Notch signaling genes with sensitivity to Notch inhibition using daptomycin [114]. In PNETs, the expression of Notch1 signaling was highly related to cancer progression [115]. The activation of Notch 1 inhibits tumor growth, suggesting that Notch 1 serves as a tumor suppressor [116,117,118]. Accordingly, Notch1 activation has been shown to induce stable disease in a fraction of PNET patients in a Phase II trial [119]. As of now, there is no documentation of DNA methylation occurring in the Notch1 signaling pathway in PNET samples.

#### 3.1.9. Wnt/β-Catenin

Wnt/β-catenin signaling is closely related to tumor progression, metastasis, migration, and invasion [120]. Wnt signaling has been known to play a large role in the pathogenesis of colon cancer, with greater than 90% of CRCs carrying a mutation that activates Wnt signaling. When Wnt signaling is activated, Wnt molecules will bind to the FZD-LRP5/6 co-receptor complex. LRP6 requires acetylation by p300 to become activated for Wnt signaling [121]. In epithelial ovarian cancer cells, alternations of DNA methylation regulate members of the Wnt/β-catenin pathway [122]. Interestingly, the combination of curcumin, commonly recognized as turmeric, with decitabine hinders the formation and migration of ovarian cancer cell colonies, suggesting that this effect might be achieved, at least partially, by the downregulation of the DNMT3a protein, leading to the inhibition of the epithelial-to-mesenchymal transition (EMT) process [123]. Adherens Junctions Associated Protein 1 (AJAP) forms a complex with E-cadherin and β-catenin in the cytomembrane, thereby diminishing the nuclear translocation of β-catenin and impeding the Wnt/β-catenin signaling pathway. In salivary adenoid cystic carcinoma (SACC) tumors, AJAP has been found downregulated, primarily due to promoter hypermethylation, which causes AJAP gene silencing. Silencing AJAP1 in SACC cells markedly amplifies the processes of proliferation, invasion, and metastasis both in vitro and in vivo. Furthermore, it independently affects the prognosis of SACC (38). Zinc-finger protein 471 (ZNF471) exerts its tumor-suppressive functions by suppressing EMT, tumor cell stemness, and the inhibition of Wnt/β-catenin signaling. In both breast cell lines and tissues, there is notable downregulation of the promoter CpG methylation of ZNF471, as observed in comparison to normal mammary epithelial cells and their corresponding surgical-margin tissues [124]. In the context of PNET development, precise regulation of the Wnt/β-catenin signaling pathway is essential for normal pancreas development [125]. On the other hand, the dysregulation of Wnt/β-catenin leads to elevated expression of β-catenin and the decreased expression of Wnt/β-catenin inhibitors, a phenomenon frequently observed in higher-grade PNETs [126]. The aberrant methylation of SFP1, an inhibitor of the Wnt/β-catenin pathway, caused the downregulation of SFP1 protein expression and enhanced tumor growth. Conversely, the overexpression of SFP1 and WIF-1, another Wnt/β-catenin inhibitor, resulted in the inhibition of tumor growth both in vitro and in vivo [127]. In addition, there is a strong correlation between promoter CpG hypermethylation and decreased expression of O6-methylguanine-methyltransferase (MGMT), a downstream target of the Wnt/β-catenin pathway, in PNETs [128,129].

#### 3.1.10. NF-κB

NF-κB signaling plays a critical role in cancer development by mediating the inflammatory tumor microenvironment [130]. There have been numerous reports of epigenetic regulation of NF-κB signaling in cancers. In colorectal carcinoma, protein arginine methyltransferase 5 (PRMT5) catalyzes methylation of the multifunctional protein Y-box binding protein 1 (YBX1) and causes NF-κB activation, which, in turn, leads to increased cell proliferation in vitro [131]. In addition, NF-κB signaling has been involved in the regulation of fatty acid-binding proteins (FABPs), the dysregulation of which has been shown in many types of cancer. FABP5 promoter hypomethylation correlates with its increased expression, which forms a positive feed-back loop with NF-κB to promote tumor metastasis [132]. On the contrary, DRD2, involved in restricting NF-κB signaling and EMT in breast cancer (BC), undergoes downregulation due to promoter hypermethylation. DRD2 demonstrates the ability to suppress tumorigenesis in both in vitro and in vivo settings. Notably, there exists a positive correlation between DRD2 expression and extended survival times in BC patients [133]. In non-small cell lung cancer (NSCLC), genetic deletion and methylation contribute to decreased expression of circadian gene hepatic leukemia factor (HLF). HLF inhibits NF-κB/p65 signaling via increasing activity of PPAR in malignant tissues [134]. In PNET samples, the upregulation of NF-kB is positively correlated with tumors with higher grades. In addition, the downregulation of NF-kB and STAT3 inhibits proliferation, viability, and spheroids growth of PNET cell lines [135]. Presently, there is no documentation on the DNA methylation status of NF-κB in PNETs.

#### 3.1.11. Somatostatin Receptor 2 (SSTR2)

Somatostatins are a family of peptide hormones that are secreted by pancreatic islet δ cells and bind to somatostatin receptor (SSTR) signaling to inhibit the release of a variety of enzymes including insulin, glucagon, pancreatic amylase, along with other hormones secreted by the pancreas. Upon the binding of SST, SSTR exerts anti-proliferative functions, thereby serving as a tumor suppressor in PNETs [79]. Since SSTR2 has been found highly expressed in a majority of neuroendocrine neoplasms (NENs) [136], the antitumor activity of SSTR2 has been explored in the treatment of NENs, including PNETs [137]. Methylation profiling of PNET samples showed hypermethylation of the SSTR2 gene promoter [138], and, not surprisingly, DNMT inhibition recovered SSTR expression in cell cultures and facilitated the delivery of novel peptide receptor radiotherapy against PNETs [139].

#### 3.1.12. SMAD3

SMAD transcriptional factors, which are part of the transforming growth factor-β (TGF-β) signaling pathway, regulate cell differentiation and proliferation [140,141]. Depending on the cell status, activated SMAD can lead to either the induction or inhibition of cell growth [141]. In PNETs, TGF-β/SMAD signaling mainly functions to inhibit cell growth via the SMAD downstream effectors, p21WAF1/CIP1 tumor suppressor [141,142]. Loss of homozygosity (LOH) of the SMAD3 gene has been observed in a fraction of PNET patients [143]. In addition, the inhibition of TGFβ/SMAD by the inhibition of SMAD7 causes over-proliferation of islet β cell proliferation in adult mice, suggesting that TGF-β/Smad signaling might play a role in PNET progression [144]. SMAD3 has been shown to have undergone promoter CpG methylation in CRC [145,146]. Nevertheless, there is currently no information available regarding the DNA methylation of SMAD3 in PNETs. A complete list of DNA methylation profiling of PNET-related pathways is shown in Table 1.

### 3.2. Histone Modifications

Histone modifications include both the acetylation and methylation of histones, which control the conformation of genomic DNA. Acetylation is a post-translational histone modification process that modulates the opening of the chromatin structure, allowing for gene transcription. Through the opposing activities of histone deacetylases (HDACs) and histone acetyl transferases (HATs), the process of acetylation is reversible through either the addition or removal of acetyl groups from the amino-terminal ε-group of lysines on histones. HDACs catalyze the removal of the acetyl moieties from acetylated histones and are generally associated with transcriptional repression [150]. Whereas histone acetylation is linked to transcriptional activation, histone methylation can either be repressive or activating. This depends on which lysine residue (K) on which histone (H3, H4) is modified and the extent of methylation, i.e., di- or tri-methylation (me2 or me3). For example, there are markers for inhibitory histone methylation such as H3K9me2/3 and H3K27me2/3. In contrast, H3K4me2/3, H2K36me3, and H3K79me3 serve as markers for activating histone methylation. The mediation of the histone methylation process involves histone lysine methyltransferases (HMTs) and histone demethylases (HDMs) [151].

Histone modifications have been shown to regulate many important biological functions, including cell cycle progression, metabolism, differentiation, and development. Not surprisingly, they are found involved in various tumors and are involved in vital chromosomal translocation-mediated oncogenic protein fusions and other carcinogenic events [152,153]. Histone modifications play a crucial role in the different stages of cancer and have been linked to a variety of malignancies, including solid and hematologic tumors. For example, increased levels of HDAC 5 have been correlated with metastatic PNETs. Similarly, elevated expressions of HDACs 1, 2, and 3 are linked to poor outcomes in gastric and ovarian cancers [154,155]. Both histone acetylation and methylation have been found involved in the regulation of multiple signaling pathways in cancer development [153].

#### 3.2.1. MEN1

Despite the lack of evidence to support a direct role for histone acetylation or deacetylation in MEN1-associated endocrine cells, there is indirect evidence through examining the effects of HDAC inhibitors (HDACis) on MEN1-associated cell lines. Menin protein regulates the cyclin B2 promoter region by modifying histone H3 acetylation and H3K4me3 methylation levels in mouse embryonic fibroblast cells (MEF) [156]. HDACi such as TSA, thailandepsin-A (TDP-A), sodium butyrate (NaB), valproic acid (VPA), and BET protein bromodomain inhibitors (BETi) have been well studied in neuroendocrine tumor cell lines including BON-1, NCI-H720, NCI-H727 and QGP-1 [157,158,159,160,161]. In addition, HDACi has been found to induce dose-dependent growth inhibition on NET cells. The anti-tumor effects of HDACis have been further demonstrated in a BON-1 xenograft model [159] and a pancreatic beta cell-specific MEN1 knockout mouse model [161]. Besides playing a role in PNET pathology, menin itself can serve as part of a mixed-lineage leukemia histone methyltransferase complex that promotes histone methylation [162]. Polycomb repressive complex 2 (PRC2)-mediated H3K27 tri-methylation regulates the genome-wide distribution of MLL1 and MEN1 in diffuse large B-cell lymphoma (DLBCL) cells. In cells with Enhancer of zeste homolog 2 (EZH2) gain-of-function mutations, combinatorial inhibition of MEN1 using MI-503 or VTP50469 with EZH2 inhibitor Tazemetostat (EPZ-6438) has been shown to decrease tumor burden and survivability in a lymphoma xenograft model [163]. In addition, aberrations of the EZH2 gene, along with MEN1 and others, can cause an imbalance in the activities of histones via methylation, contributing to parathyroid tumorigenesis in parathyroid carcinoma [164]. MEN1 even plays a role as a tumor suppressor gene in CRC, with recent work finding a hotspot mutation in MEN1 that affects 4% of BRAF mutant cancers [165].

#### 3.2.2. mTOR-TSC

In glioblastoma, members of the mTOR-TSC pathway have been found to be acetylated, resulting in the amplification of c-Myc and the promotion of glutaminolysis. mTORC2 regulates c-Myc levels through Akt-independent phosphorylation of class II HDACs, which leads to the acetylation of FoxO1 and FoxO3 and the release of c-Myc (4,5,7,9). The upregulation of mTORC2, c-Myc, and acetylated FoxO are correlated with worse prognosis in glioblastoma patients. The pharmacologic inhibition of PI3K and mTOR kinase has been found to suppress FoxO acetylation, decrease c-Myc levels, and result in tumor cell death [166]. mTORC1 and mTORC2 cooperatively cause hypermethylation of H3K27, which subsequently promotes tumor cell survival of glioblastoma both in vitro and in vivo, suggesting that dynamic regulation of histone methylation by mTOR complexes could be exploitable as a novel therapeutic target against this deadly tumor [167]. PTEN has been found to be acetylated at K125 and K128 by the p300/CBP associated factor (PCAF), resulting in its inactivation and upregulation of the PI3K-AKT signaling pathway [168]. TSA and SAHA, another HDAC inhibitor, up-regulate PTEN by inducing its membrane translocation through acetylation at K163 in 293T cells. Further studies have determined that HDAC6 inhibition, using the HDAC6-specific inhibitor tubastatin A, caused PTEN acetylation and activation [169]. There is currently no report of histone modifications of the mTOR pathway regarding PNETs.

#### 3.2.3. HIF1α-VHL

Epigenetic regulation of HIF1α signaling has been well-established in a variety of cancers. P300 is a transcriptional component of HIF1α, which acetylates HIF1α at Lys-709, increasing its stability as shown in both osteosarcoma and renal cell carcinoma cell lines [170]. The HDAC inhibitor SAHA has been found to decrease HIF1α levels in tumor cell lines via direct acetylation of heat shock protein 90 (Hip90), a HIFα chaperone protein, using HDAC6 [171]. In addition, HIF1α R282 has been shown to be methylated by protein arginine methyltransferases 3 (PRMT3) in CRC, which is necessary for its stabilization and oncogene function. PRMT3 inhibition with a novel therapeutic molecule (MPG-peptide) decreases tumor growth and angiogenesis in vivo, potentially offering a future therapeutic strategy [172]. Presently, there is no available information on the histone modifications of the mTOR pathway regarding the HIF1α-VHL signaling in PNETs.

#### 3.2.4. RAS-MAPK-NF1

RAS has been found acetylated in human cancer lines on lysine 104, resulting in destabilization of its Switch II domain, which modulates the interaction between RAS and GEF as well as RAS and PI3K for GAP-induced GTP hydrolysis. Essentially, acetylation stands as a negative regulator of RAS function, exerting anti-oncogenic effects [173]. Other studies have established a link between RAS signaling and H3K9ac modification, which is one of the most widely studied acetylation sites of histone H3 tails and has been commonly observed in ovarian cancer, hepatocellular carcinoma (HCC), oral cancer, and cervical cancer [174]. Through binding to xeroderma pigmentosum A protein (XPA), RASSF1A regulates the acetylation and deacetylation cycle of XPA [175]. Not only does RASSF1A act as a tumor suppressor through DNA repair, but RASSF1A has also been found to increase microtubule stability through HDAC6, which prevents tumor cell migration, invasion, and metastasis [69]. The effect of RASSF1A was also demonstrated in the NSCLC xenograft model and human bronchial cells [176]. Patients with NF1 familial syndrome sometimes develop benign plexiform neurofibromas, which can develop to form peripheral nerve sarcomas known as malignant peripheral sheath tumors (MPNSTs). Bromodomain containing protein 4 (BRD4), a histone acetyltransferase, binds to acetylated histone H3K27Ac in MPNSTs and has been found to be one of the most highly unregulated genes in MPNSTs. The Bromodomain-containing inhibitor, JQ-1, has been demonstrated to decrease MPNST tumor growth in vitro and in vivo [177]. A phase II trial was conducted in patients with MPNST using JQ-1 (NCT02986919); however, the trial was discontinued due to a lack of participants. Currently, there is a new Phase 1/2 Trial that is evaluating the efficacy of the Bromodomain-containing inhibitor AZD5153 in combination with PL1 therapy and MEK inhibitor therapy for the treatment of neurofibromatosis [178]. In PNETs, the only report of histone modifications in RAS-MAPK signaling came from one of the receptor tyrosine kinases, IGF2, which showed aberrant H3K4 methylation in the pancreatic islets of MEN1-deficient mice [179].

#### 3.2.5. ATRX/DAXX

The chromatin remodeler ATRX and the histone chaperone DAXX are mutated in a variety of cancers, including glioblastoma multiforme, pediatric adrenocortical carcinoma, osteosarcoma, neuroblastoma, myelodysplastic syndrome, acute myeloid leukemia, and PNETs [180]. As mentioned in Section 3.1, ATRX and DAXX form a histone chaperone complex that deposits histone variant H3.3 into specific genomes. The first discovery of ATRX/DAXX mutations came from PNETs, where they were found to be mutated in 43% of the tumors. These mutations are inactivating mutations and exhibit alternative lengthening of telomeres (ALT) phenotype [5]. DAXX has been found to exist in two distinct H3.3 complexes, which include the DAXX-ATRX complex and the DAXX-SETB1-KAP1-HDAC1 complex [181]. The DAXX-SETB1-KAP1-HDAC1 represses endogenous retroviral (ERV) sequences. ERVs lead to genomic instability via insertion into protein-coding regions or by promoting the transcription of neighboring genes, leading to tumorigenesis [182].

#### 3.2.6. CDKN2A-RB1

RB1 antagonizes E2F activity through binding to E2F transcription factors and multiple corepressor molecules, including HDACs (HDAC1, 2, 3) [183,184]. HDACis, such as SAHA, have been shown to cause RB1-mediated silencing of cell cycle oncogenes [185]. More recently, HDAC5 has been found to act as a corepressor for RB1 but can be inhibited by CDK 4/6. CDK 4/6 inhibition was found to enhance specifically HDAC5 binding to RB1. Resistance to CKD 4/6 inhibitor therapy, such as palbociclib, was found to be related to low HDAC levels in cancer cells, which means HDAC5 could act as a useful biomarker in guiding therapy [186]. HDAC inhibition has been shown to have anti-proliferative effects by inducing cell cycle arrest in G1 via cycling-dependent kinase inhibitors or the downregulation of cyclins and CDKs [187]. Inhibition of HDACs induced elevated expression of CDK inhibitors, resulting in the blockade of the cell cycle at the G1 phase and disruption of the G1/S transition. This occurred through the reactivation of RB1 function, leading to the inhibition of E2F, halt of G1 progression, and cell apoptosis [188,189]. HDACis such as TSA, SAHA, and VPA have been shown to result in cell cycle arrest at the G2/M checkpoint [190]. HDACs not only play a transcriptional role in cell cycle progression but can also act as a cell cycle regulator by modulating Aurora B kinase in mitotic progression [191]. Maggi et al. reported that retinoblastoma binding protein 2 (Rbp2), a histone demethylase, was found overexpressed in PNETs. In addition, aberrant expression of Rbp2 altered histone demethylation and contributed to PNET pathogenesis [192].

#### 3.2.7. P53

One of the first recognized non-histone proteins affected by acetylation was p53 [193,194]. P53 is a tumor suppressor that is negatively regulated through MDM2 and transiently stabilized and activated by various intracellular processes. While the phosphorylation of serine residues was previously recognized as the primary stabilizer of p53, it has been discovered that acetylation also plays a role in stabilizing p53 during cellular stress and contributes to enhancing its transcriptional activity [194]. MDM2 has been reported to negatively regulate p53 acetylation but can be reversed by tumor suppressor p14^ARF^ [194]. In H1299 carcinoma cells, acetylation status regulated the degradation of p53 by directly interacting with MDM2 E3 ligase [195]. HDACis, such as TSA and SIRT1, were found to reverse the inhibitory effects of cAMP on DNA damage-induced p53 stabilization and apoptosis [196]. TSA has also been found to upregulate the binding of PUMA promoter by p53 in GC through inhibition of HDAC3 [197]. Recent evidence has found that p53 itself can be a driver of HDAC inhibition, which in turn leads to autophagy and programmed cell death [198]. VPA and TSA were found to induce apoptosis in Panc1 and PaCa44, two pancreatic cancer cell lines, through the upregulation of p21 and PUMA, resulting in mutant p53 degradation [199]. Currently, there is no available information on histone modifications of p53 in PNET research.

#### 3.2.8. Notch Signaling

Upon activation, NICD forms a coactivator complex that includes RBPJ and histone acetyltransferase p300. During this process, HDAC3 increases NICD stability by reducing the p300-mediated acetylation of NICD protein. HDAC3 expression was found to be significantly higher in T-ALL patients and CLL B cell samples [110]. In patients with breast cancer, elevated expression of histone methyltransferase NSD3 was associated with recurrence, distant metastasis, and poor survival. In mice, NSD3 promoted malignant transformation of mammary epithelial cells [200]. NSD3-induced methylation of H3K36 is crucial for Notch-dependent tumor initiation and metastasis, suggesting that Notch-related methylation factors may be an actionable therapeutic target in localized and metastatic breast cancer [201]. VPA has been found to activate Notch signaling in neuroendocrine tumor cells, resulting in the suppressed growth of carcinoid tumors in a mouse tumor xenograft [116,202]. These findings were further confirmed in a phase II pilot study that examined the effects of VPA on eight patients with low-grade neuroendocrine tumors (carcinoid and pancreatic). Pretreatment tumor biopsies revealed low Notch1 levels and a 10-fold increase in Notch1 activity on a post-VPA treatment tumor biopsy [119].

#### 3.2.9. Wnt/β-Catenin

Histone modifications have been shown to be involved in the regulation of Wnt signaling via complicated mechanisms. Upon activation, Wnt molecules bind to the FZD-LRP5/6 co-receptor complex. LRP6 requires acetylation by p300 to become activated for Wnt signaling. When Wnt is activated, β-catenin is released from the destruction complex and can become acetylated by CBP, p300, and PCAF, which allows for the upregulation of β-catenin activity. Β-catenin will then undergo nuclear translocation to activate gene transcription [121]. When Wnt signaling is in its inactive state, beta-catenin becomes phosphorylated by the protein destruction complex and targeted for proteasomal degradation. Once degraded, a repressive complex containing a T cell factor (TCF)/lymphoid enhancer factor (LEF) and a transducing-like enhancer protein (TLE) utilizes HDACs to repress target genes [203]. HDACis such as TSA have been found to inhibit the Wnt/β-catenin pathway by increasing β-catenin phosphorylation, reducing β-catenin nuclear translocation, and inhibiting β-catenin/TCF complex formation in pituitary corticotrope tumor cells (AtT20) [204]. TSA and SAHA have demonstrated anti-tumor effects in colon cancer cell lines with β-catenin mutations by downregulating the Wnt transcription factor TCF7L2 [205]. Curcumin, as discussed in Section 3.1, also serves as an inhibitor of acetyltransferase, specifically targeting p300 in the Wnt pathway. Currently, curcumin is undergoing Phase I clinical trials for the treatment of breast, colon, and pancreatic cancers [206]. Currently, there is no available information on histone modifications of Wnt/β-catenin in PNET research.

#### 3.2.10. NFκB

Histone modifications play a major role in the regulation of NF-KB signaling. These modifications occur in the cytoplasm and allow for the activation of IκB kinases (IKKs), resulting in the degradation of IκBα, which allows for translocation of NFκB into the nucleus [207]. Within the nucleus, Re1A, a subunit of NFκB, undergoes acetylation by p300/CBP at lysine 310, which promotes transcriptional activity of NFκB [208]. Stat 3 is an oncogenic transcription factor that is commonly activated in cancer. Stat 3 has been found to maintain tumor NFκB activity through acetylation of Re1A. This has been studied in human A2058 melanoma cell lines and DU145 prostate cancer cells [209]. To date, there have been no documented instances of histone modifications affecting NFκB in PNETs.

#### 3.2.11. SSTR2

In the PNET cell lines BON-1 and QGP-1, histone 3 acetylation was observed on SSTR2 [210]. This finding was subsequently corroborated by Veenstra et al. [211], suggesting that the regulation of SSTR2 expression is likely influenced by both histone acetylation and DNA methylation [138]. In addition, using an in vivo BON-1 xenograft mouse model, Kidd et al. showed that the combination treatment of HDACi (VPA) and camptothecin-somatostatin conjugate significantly reduced growth compared to monotherapies, respectively [212], suggesting that combinatorial therapy might hold promise in PNET patient treatment.

#### 3.2.12. SMAD3

Epigenetic regulation of SMAD3 via histone modifications plays important roles in cancer progression and metastasis [213]. In lung cancer, the actin-binding protein profilin-2 binds to and thereby prevents HDAC1′s access to the promoters of SMAD2 and SMAD3, which causes activation and promotes EMT and angiogenesis in lung cancer cells [214]. Besides acetylation, histone methylation has also been observed in lung cancer cells. SMAD3 mediates the recruitment of the histone methyltransferase SETDB1, which controls the expression of SNAIL1 and EMT in breast cancer [215]. Despite its crucial regulatory role in PNETs, as outlined in Section 3.1, there is presently no information on histone modifications of SMAD3 in the pathology of PNETs. A complete list of histone modifications in PNET-related signaling pathways is listed in Table 2.

### 3.3. Non-Coding RNAs

Alterations in both coding and ncRNAs have been widely implicated in cancer pathophysiology. The ncRNAs primarily consist of miRNAs and lncRNAs. MiRNAs are characterized by the lengths of 18–22 nucleotides, whereas lncRNAs encompass 200 nucleotides or more [220]. There are a few recent reviews that described the limited literature concerning the role of ncRNAs in pancreatic adenocarcinoma cancers and PNETs [221,222,223]. However, compared to other heavily studied cancers, there is still a lack of study of miRNAs and long non-coding RNAs (lncRNAs) due to the low incidence of PNET cases.

#### 3.3.1. MEN1

NcRNAs exert nuanced control over gene expression, contributing to the dynamic modulation of the MEN1 pathway in cancer progression. For instance, multiple miRNAs, including miR-142-3p, let-7i, miR-125a-5p, miR-199b-5p, and miR-1274b_v16.0, and miR-193b have been found differentially expressed in MEN1 parathyroid tumors comparing to normal parathyroid tissues [224]. Furthermore, lncRNA NEAT1 has been implicated in the dysregulation of MEN1 signaling in multiple endocrine neoplasia type 1 (MEN1) syndrome, acting as a competing endogenous RNA to sequester miR-34a and alleviate its inhibitory effect on MEN1 [32,225,226]. Additionally, miR-29a has been identified as a negative regulator of MEN1 in parathyroid adenomas, influencing cell proliferation and apoptosis [227]. These examples underscore the intricate interplay between ncRNAs and the MEN1 pathway, highlighting the diverse regulatory roles of ncRNAs in shaping the landscape of cancer progression. Investigating these molecular interactions offers promising avenues for understanding the underlying mechanisms and identifying novel therapeutic targets in MEN1-associated cancers. Human maternally expressed gene 3 (MEG3) encodes an RNA transcript that exhibits tumor suppressive ncRNA functions (lncRNA MEG3) with an unknown contribution to cellular processes in its limited translated form [228,229,230]. MEG3 levels are downregulated in PNETs through a variety of mechanisms. In murine embryonic stem cells undergoing differentiation into pancreatic islet-like endocrine cells (PILECs), a biallelic MEN1 mutation resulted in reduced menin protein levels, subsequently leading to decreased transcription of MEG3 [179]. In both human PNET samples and MIN6 murine insulinoma cells, the hypermethylation of the MEG3 promoter is associated with the downregulation of MEG3 compared to normal tissue [231]. Luzi et al. used a BON-1 luciferase reporter cell line to show that menin acted as a negative regulator of miR-24-1, which in turn downregulated menin in a negative feedback loop [232]. In a subsequent investigation, it was demonstrated that reduced menin expression resulting from mutations in neuroendocrine cancers disrupted the feedback loop between menin and miR-24-1. This disruption further contributed to a decrease in menin levels due to the unregulated expression of miR-24-1 [233]. These studies unequivocally demonstrate that diminished menin levels, with miR-24-1 as a driving factor, play a role in the pathogenesis and progression of NENs [234].

#### 3.3.2. mTOR-TSC

Many ncRNAs have been found to modulate mTOR signaling, influencing various facets of cancer progression. In HCC, miR-99a has been identified as a critical regulator by directly targeting mTOR, suppressing its expression, and impeding cell proliferation [235]. In breast cancer, the lncRNA HOTAIR acts as a molecular scaffold to facilitate the assembly of the mTOR signaling complex, promoting tumor growth [236]. Additionally, the lncRNA TUG1 has been shown to modulate mTOR signaling in renal cell carcinoma, influencing cellular processes such as proliferation and apoptosis [237]. In the MIN6 luciferase reporter cell line, an overexpression of miR-144 downregulated PTEN, which in turn caused the upregulation of the p-AKT pathway and a subsequent increase in beta cell proliferation [238]. This piece of evidence is in line with the expression pattern of p-AKT and miR-144 in a differential expression analysis of 29 human insulinoma samples in comparison with normal tissues [238]. In addition, overexpression of a lncRNA H19 plasmid activated VGF and PI3K/AKT pathway in the QGP-1 cell line, suggesting a link between lncRNA H19 and an increase in proliferation and metastasis of neuroendocrine tumors [239].

#### 3.3.3. HIF1α-VHL

Hypoxia-responsive non-coding RNAs are crucially involved in controlling gene expression under hypoxic conditions, operating at various stages such as transcriptional, posttranscriptional, translational, and posttranslational levels, influencing critical aspects of cancer progression. Notably, miR-210 has been identified as a key regulator by directly targeting HIF-1α, promoting angiogenesis and metastasis in various cancers [237,240]. In renal cell carcinoma, the lncRNA H19 acts as a molecular sponge for miR-29a, derepressing HIF-1α expression and fostering tumor growth [241]. Additionally, miR-20a has been implicated in the regulation of HIF-1α in CRC, influencing cellular responses to hypoxia [242]. The lncRNA MALAT1 has also been associated with HIF-1α modulation in breast cancer, affecting tumor invasion and metastasis [243]. These examples underscore the intricate roles of ncRNAs in shaping the dynamics of HIF-1α signaling in cancers. There are a few reports regarding miRNA-mediated regulation of HIF1α signaling in PNETs. Thorns et al. compared the differential expression of miRNAs in 37 PNET tissue samples and nine non-neoplastic controls, from which they were able to identify two miRNAs. While levels of miR-642 were positively correlated with Ki-67 scores, levels of miR-210 expression were positively correlated with metastasis [244]. The connection between miR-210 and its targets has not been elucidated in PNETs, although in liver metastasis samples from colorectal adenocarcinoma patients, miR-210 was shown to affect disease progression possibly through interaction with the HIFa pathway [245,246].

#### 3.3.4. RAS-MAPK-NF1

The Ras-MAPK signaling pathway plays a pivotal role in regulating various cellular processes, and the influence of ncRNAs on this pathway has emerged as a significant area of exploration in cancer research. For instance, miR-21 has been implicated in the activation of Ras-MAPK signaling in pancreatic cancer, promoting cell proliferation and invasion [247]. Conversely, miR-143 functions as a tumor suppressor by targeting KRAS, a key component of the Ras pathway, and inhibiting its downstream signaling in CRC [248]. In melanoma, the lncRNA SPRY4-IT1 has been shown to activate the Ras-MAPK pathway by regulating SPRY4 expression [249]. Additionally, the lncRNA MALAT1 has been associated with Ras-MAPK signaling in lung cancer, influencing cell proliferation and migration [250]. Using both a QGP-1 GFP reporter cell line and a murine xenograft model, Zhang et al. showed that miR-431 promoted EMT, invasion, and metastasis by silencing DAB2IP, a tumor suppressor protein whose downregulation resulted in the direct activation of the Ras pathway [251].

#### 3.3.5. DAXX/ATRX

In glioma, miR-1269a has been identified as a regulator that targets ATRX, promoting cell proliferation and invasion [252]. In addition, miR-21 is found upregulated in GC. Accordingly, overexpression of miR-21 promotes tumor growth by targeting DAXX in GC cell lines [253]. Furthermore, the lncRNA XIST has been implicated in the regulation of DAXX-ATRX in glioblastoma, influencing both chromatin remodeling and tumor progression [254]. In the context of PNET research, Gill et al. performed microarray differential expression profiling of miRNA in 37 PNET tissue samples, including local and distant metastases, and found a significant upregulation of miR-3653 in the metastatic group. Following bioinformatic analysis of three miRNA databases, ATRX was revealed to be a negatively regulated target for miR-3653, suggesting a possible link between miR-3653 and PNET pathogenesis through the dysregulation of the DAXX/ATRX pathway [255].

#### 3.3.6. CDKN2A-RB1

Recent investigations reveal that ncRNAs play intricate roles in modulating RB1 signaling, contributing to the complex landscape of cancer progression. For instance, miR-106b-5p has been identified as an oncogenic miRNA that targets RB1 signaling, promoting cell cycle progression and proliferation in laryngeal carcinoma [256]. Conversely, miR-34a and miR-15a/16 act synergistically to induce cell cycle arrest and apoptosis in NSCLC cell lines in an RB1-dependent manner [257]. In addition, the H19 ncRNA controls cell proliferation and metastasis by regulating essential RB1-E2F signaling in CRC [258]. Furthermore, the lncRNA PVT1 has been shown to regulate RB1 in GC, contributing to tumor growth [259]. Despite its critical regulatory role in PNETs, as described in Section 3.1, there have been no documented instances of ncRNAs affecting RB1 in PNETs.

#### 3.3.7. p53

As a pivotal sequence-specific transcription factor, p53 plays a vital role in governing the expression of numerous genes, including both miRNAs and lncRNAs. The exploration of p53-associated lncRNAs has proven highly promising, showing potential as biomarkers for cancer diagnosis or as targets for disease therapy [260]. For example, in leukemia, miR-125b has been implicated as a cancer-associated miRNA (oncomiR) that targets p53, promotes cell proliferation, and inhibits apoptosis [261]. On the other hand, miR-34a, a well-established tumor suppressor, directly targets p53, inducing cell cycle arrest and apoptosis in multiple cancer types, including breast and lung cancers [262,263]. The lncRNA TP53TG1 also acts as a positive regulator of p53 by enhancing its stability and transcriptional activity in GC [264]. Additionally, the NEAT1 functions as a tumor suppressor in HCC in a p53-dependent manner [265]. Furthermore, Zhou et al. showed that MEG3 caused the downregulation of MDM2 expression and stabilized p53, suggesting that lncRNA MEG3 stimulates p53 transcription and is linked to p53 longevity and activity [266]. As of now, there is no available information on the regulation of p53 signaling in PNETs through ncRNA mechanisms.

#### 3.3.8. Notch

NcRNAs can participate in the NOTCH signaling pathway by serving as regulators of various target genes [267]. For instance, the expression of Notch-1 and miR-21 were found to have a positive correlation with the development of CRC, suggesting that miR-21 may function to promote cancer progression [268]. Conversely, miR-34a has been identified as a key player in negatively regulating Notch signaling by targeting Notch1 and Jagged1, leading to suppressed proliferation and enhanced apoptosis in CRC [269]. In addition, miR-34a has been demonstrated to directly inhibit Notch-1/2 by binding directly to their 3’ UTR in glioma cells [270]. Moreover, the lncRNA HOTAIR regulates the expression of notch3 by competitively binding with miR-613, making the notch3-HOTAIR-miR-613 complex a potential drug target in pancreatic cancer [271]. In the context of PNETs, by combining RNASeq and IHC expression analysis, He et al. showed that the upregulation of lncRNA XLOC_221242 correlates with overexpression of DNER protein and activation of a variety of precursors in the Notch and Wnt signaling pathways, suggesting that lncRNA XLOC_221242 might interact with Notch signaling in the regulation of PNET progression [272].

#### 3.3.9. Wnt/β-Catenin

The interplay between non-coding RNAs (ncRNAs) and the Wnt/β-catenin signaling pathway plays a crucial role in determining tumorigenesis. Elevated Wnt expression is often observed in cancers, and microRNAs (miRNAs) can bind to the 3′-UTR of Wnt, leading to a reduction in its levels. For instance, the tumor suppressor miR-34a directly targets Wnt/β-catenin pathway components, inhibiting cell growth and inducing apoptosis in HCC [273]. Conversely, in CRC, miR-21 has been identified as an oncomiR that promotes Wnt/β-catenin signaling activation by targeting negative regulators, contributing to enhanced cell proliferation and invasion [274]. The lncRNA CCAT2 forms a feedback loop with Wnt/β-catenin signaling in colon cancer [275]. In the context of PNET research, a decrease in MEG3 caused the upregulation of miR-183 in BON-1 cells. MiR-183 functions as a downstream target for MEG3 and is negatively regulated by MEG3. When miR-183 activity was increased, it upregulates BRI3, whose positive relationship correlates with p38/ERK/AKT and Wnt/β-catenin pathways, suggesting a complicated tumor-promoting mechanism which includes interaction among MEG3, miR-183, and BRI3 via p38/ERK/AKT and Wnt/β-catenin signaling [230,231]. In addition, in BON-1 cells and LCC-18 colonic neuroendocrine cell lines, overexpression of lncNEN885 caused a decrease in invasion and EMT by downregulating Wnt/β-catenin signaling. siRNA silencing of lncNEN885 restored markers of EMT and a component of the Wnt/β-catenin pathway, suggesting that lncNEN885 might contribute to the regulation of neuroendocrine tumor metastasis through the Wnt/β-catenin pathway [276].

#### 3.3.10. NFκB

The NFκB signaling pathway, a central regulator of inflammation, immunity, and cell survival, is intricately modulated by ncRNAs in the context of cancer [277]. For example, in breast cancer, miR-1892b has been identified as a key negative regulator of NFκB, thereby inhibiting inflammatory responses and tumor growth [278]. On the other hand, the oncomiR miR-21 promotes NFκB activation by targeting negative regulators in colon adenocarcinomas, as shown in evidence from diverse cell lines, xenograft mouse models, and gene expression profiling of patient tumor samples [279]. Additionally, the lncRNA MALAT1 has been associated with NFκB signaling in oral squamous cell carcinoma [280]. Furthermore, the lncRNA NEAT1 inhibits miRNA-216b and promotes CRC progression [281]. In the context of PNETs, Huang et al. studied the differential expression of miRNAs in 37 PNET samples and compared the differences between localized and metastatic tumors. Their findings indicated that miR-196a is involved in NFκB signaling and exhibits a notable correlation with elevated tumor grade, advanced stage, and tissue invasion in PNETs [282].

#### 3.3.11. SSTR2

To date, there have been limited reports on the interplay between ncRNAs and SSTR2 signaling in cancers, with the existing literature mainly focusing on neuroendocrine neoplasms. For instance, the combination of SSTR2 with miR-7 and miR-148a caused inhibition of the growth of both lung and intestinal carcinoid cell lines [283]. In PNETs, it has been reported that the combination of overexpression of miR-16-5p and the somatostatin analog octreotide resulted in the upregulation of SSTR2 expression in the INS-1 neuroendocrine cell line. This suggests that miRNAs may play a role in advancing the development of combinatorial therapeutic approaches, particularly for patients who do not respond adequately to somatostatin analog treatment targeting SSTR2 alone [284].

#### 3.3.12. SMAD3

NcRNAs interact with TGF-β/SMAD3 signaling to regulate important facets of cancer cell development, including EMT, invasion, migration, cancer cell stemness, and metastasis [285]. For example, miR-21 targets Smad7, an inhibitor of TGF-β/SMAD3 signaling, which leads to enhanced cell proliferation and invasion in various cancers [286]. Conversely, the tumor-suppressive miR-145 directly targets SMAD3, inhibiting TGF-β-induced EMT and metastasis in CRC [118]. In addition, miR-15a correlates with SMAD3 expression in NSCLC tissues. Not surprisingly, the overexpression of miR-15a caused a significant downregulation of SMAD3 expression and inhibited the proliferation of the A549 lung cancer cell line [287]. Furthermore, lncRNA H19 has been found associated with the regulation of TGF-β/SMAD3 in HCC, impacting tumor growth [288]. Despite its important regulatory role in PNET pathology, as described in Section 3.1, as of now, there are no reports on the miRNA-dependent regulation of SMAD3 in research studies focusing on PNETs. A complete list of ncRNAs in PNET-related signaling pathways is listed in Table 3.

## 4. Future Directions for Epigenetic Research and Clinical Applications in PNET Patient Care

Up to this point, we have explored the epigenetic regulation of signaling pathways in the context of both PNETs and other extensively studied cancers. It is important to note that our intention is not to present exhaustive reviews that include detailed introductions to all the implicated signaling pathways. We acknowledge that providing such an in-depth review of the epigenetic regulation of even a single signaling pathway, like MEN1 signaling, would necessitate a comprehensive review in its own right [234,292,293]. Therefore, rather than aiming to cover all signaling pathways extensively, our focus is on discerning the disparities or gaps in our understanding of the epigenetic regulation of signaling pathways between PNETs and other extensively studied cancers. The overarching goal is to utilize the understanding derived from existing knowledge in various cancers and translate this knowledge into advancements in research and clinical applications specific to PNETs. This is particularly pertinent in the pursuit of more effective epigenetic drugs, akin to the successful cases observed in extensively studied cancers.

There has been notable advancement in comprehending the epigenetic control of cancers, although the available FDA-approved epigenetic therapies remain relatively scarce, with the majority targeting blood-borne cancers. Importantly, none of these approved therapies are directed at PNETs. Although there have been endeavors to explore the use of FDA-approved drugs in clinical trials for PNET treatment, no successful reports have emerged thus far, as detailed in Table 4.

It is worth noting that currently approved FDA epigenetic drugs, such as DNMTi and HDACi (Table 4), lack pathway specificity, thereby causing severe off-target effects and hindering their broad applications in the ever-growing field of precision medicine. In response to this limitation, ongoing efforts involve the utilization of CRISPR-guided systems to selectively activate or silence the promoter regions of either tumor suppressor or oncogenes, respectively [299]. However, this approach necessitates a more comprehensive understanding of the epigenetic regulation governing signaling pathways in PNETs.

As outlined in Section 3, it is evident that the epigenetic regulation of PNETs remains an insufficiently explored domain in comparison to other extensively researched cancers, barring a few exceptions like the MEN1 and SSTR2 signaling pathways. This discrepancy is noticeable across various facets of epigenetic research, spanning from comprehensive genome profiling of PNET patient samples to laboratory-based animal studies and gene manipulation of cell lines (refer to Table 1, Table 2 and Table 3). For instance, in the field of DNA methylation research, there are a lack of large-scale epigenetic profiling of PNET patient samples. As of 2022, there have been only nine studies reporting global DNA methylation in human PNETs, which includes a total of 739 samples [300]. The lack of research samples could be, at least partially, due to the lack of overall PNET patient cases [1]. As discussed in Section 1, PNETs make up around 2% of all pancreatic malignancies, characterized by their slow-growth tendencies and metastatic potential. These special characteristics resulted in the limited availability of patient tissue samples, particularly in the case of high-grade PNETs obtained from tumor autopsies [1]. Larger-scale studies and more advanced technologies are desperately needed to validate the existing results because the identification of molecularly different NET subtypes will have a significant impact on clinical practice, given the high heterogeneity of PNETs. For example, in the research of some commonly occurring cancers, there have been reports of using advanced single-cell whole genome methylation profiling to obtain much more complicated information about the cancer epigenome [301].

The unique slowing growth characteristic of PNETs also presents a practical challenge for research work in laboratories. For instance, STC-1, one of the most used mouse PNET cell lines, has a doubling time of 54 h, which not only makes it hard to perform gene manipulation but also creates a practical hurdle for creating syngenetic xenograft mouse models [302], which have been proven particularly suitable for studies of tumor immunity and immunotherapy response because of the presence of fully functional murine immune system [303]. Another challenge lies in the scarcity of PNET modeling systems. Currently, only a handful of human PNET cell lines, including the most widely used BON-1 and QGP-1, are accessible for research purposes. Consequently, there is a pressing need for further efforts to enhance and diversify the available tools for verifying both gain-of-function and loss-of-function in the continuously expanding list of genes associated with the epigenetics research of PNETs [304].

Despite the inherent challenges, some encouraging advancements have been achieved. Notably, recognizing the limited effectiveness of epigenetic cancer drugs has prompted investigations into combining these therapies with existing or emerging targeted treatments for a synergistic and combinatorial approach. This innovative strategy holds promise for improving clinical outcomes. For example, combination therapies that concurrently enhance SSTR expression using HDACis and target SSTRs may demonstrate enhanced efficacy compared to individual therapies [17] (also refer to Section 3.2). Additionally, our laboratory has previously demonstrated that a combination of 5-azacytidine and chemotherapy can effectively reduce cell proliferation, activate silenced tumor suppressor genes, and diminish tumors in vivo [305].

Moreover, within the spectrum of diverse biomarkers, miRNAs display characteristics such as stability, relative abundance, and accessibility in blood samples. These qualities make miRNAs highly suitable as biomarkers for the diagnosis and monitoring of cancer [306,307]. Consequently, there is a call for more extensive global miRNA sequencing of PNET samples to establish a more dependable diagnostic platform for the optimal care of patients with PNETs.

## 5. Conclusions

In summary, the epigenetic regulation of PNETs exhibits similarities with broader cancer scenarios, providing valuable insights and therapeutic possibilities. The ongoing exploration of the specific mechanisms governing epigenetic changes in PNETs, alongside innovative strategies for drug development, holds considerable promise for enhancing the prognosis and quality of life for individuals grappling with this challenging malignancy. Notably, epigenetic changes play pivotal roles in immune surveillance and the development of drug resistance. As a result, the use of epigenetic drugs, including inhibitors targeting various enzymes like DNMTs, HMTs, HDMs, HATs, and HDACs, has the potential to effectively complement other treatments, such as standard chemotherapy or immunotherapy, in the care of PNET patients.

## Figures and Tables

**Figure 1 ijms-25-01331-f001:**
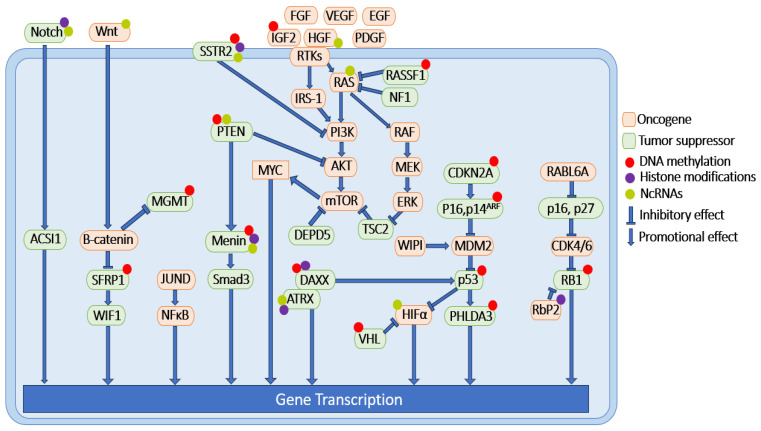
Schematic of epigenetic modifications of signaling pathways in PNET pathogenesis. Pathways are color-coded based on whether they are tumor-suppressor or oncogenic. Additional colors were added to indicate whether the regulatory mechanism of the signal molecules belongs to DNA methylation, histone modifications, or ncRNAs in PNETs. For the enhancement of clarity and ease of understanding, specific protein molecules were organized with canonical pathways, even when these molecules are not typically mentioned alongside the canonical pathways. One such example is the inclusion of TSC in the mTOR pathway based on correlations identified in previously published studies [30]. Abbreviations: Acyl-CoA synthetase isoform 1(ACSL1); Secreted frizzled-related protein 1 (SFRD1); WNT Inhibitory Factor 1(WIF1); Phosphatase and tensin homolog (PTEN); Receptor tyrosine kinase (RTK); Insulin receptor substrate 1 (IRS1); DEP domain containing 5 (DEPDC5); Hypoxia-inducible factor 1 α (HIF1 α); Cyclin-dependent kinase inhibitor 2A (CDKN2A); Wild-type p53-induced phosphatase 1 (WIP1); Mouse double minute 2 (MDM2); Pleckstrin Homology Like Domain Family A Member 3 (PHLDA3); Rab-like protein 6 (RABL6); and Cyclin-dependent kinase 4 and cyclin-dependent kinase 6 (CDK4/6).

**Table 1 ijms-25-01331-t001:** DNA methylation modifications in PNETs. ^1^ RTKs include IGF, FGF, VEGF, EGF, HGF, and PDGF.

Signaling Pathway	Signaling Molecules	DNA Methylation Status of Promoter Region	Experimental Systems
MEN1	MEN1	Hypermethylated	human PNET samples [147]
PTEN/PI3K/AKT/ mTOR/c-Myc/TSC/ RTK ^1^	PTEN	Hypermethylated	human PNET samples [52]
	TSC	no change	human PNET samples [52]
	IGF2	hypermethylated	human PNET samples [148]
HIF-1α/VHL	VHL	Hypermethylated	human PNET samples [61]
RAS/MAPK/NF1	RASSF1	Hypermethylated	human PNET samples [80]
ALT/DAXX/ATRX	DAXX	Hypermethylated	human PNET samples [88]
CDKN2A/RB1	CDKN2A	Hypermethylated	human PNET samples [98]
	P16, P14^ARF^	Hypermethylated	human PNET samples [149]
	P27	no change	human PNET samples [100]
	RB1	Hypermethylated	human PNET samples [52]
P53	P53	Hypermethylated	human PNET samples [52]
	PHLDA3	Hypermethylated	human PNET samples [108]
Wnt/β Catenin/ MGMT	SFRP1	Hypermethylated	BON-1 and QGP-1 cell lines [127]
	WIF1	no change	BON-1 and QGP-1 cell lines [127]
	MGMT	The MGMT-promoter methylation status correlates with chemoresistance in well-differentiated PNET.	PNET patient samples [128,129]
SSTR	SSTR2	The SSTR2 promoter is hypermethylated in PNETs compared to non-NET tissue and is inversely correlated with SSTR2 protein expression.	human PNET samples BON-1 and QGP-1 cell lines xenograft mouse model [139]

**Table 2 ijms-25-01331-t002:** Histone modification status in PNETs. ^1^ RTKs include IGF, FGF, VEGF, EGF, HGF, and PDGF.

Signaling Pathway	Signaling Molecules	Histone Modification Status	Experimental Systems
MEN1	MEN1	Loss of menin causes H3K4me3 loss and sporadic PNETs.	PNET patient samples [179]
PTEN/PI3K/ AKT/mTOR/ c-Myc/TSC/ RTK ^1^	IGF2	Genome-wide studies of H3K4 methylation in pancreatic islets indicate that loss of MEN1 alters the epigenetic landscape of its target genes, such as insulin-like growth factor binding protein 2 (Igf2bp2), p18ink4c (CDKN2C) and p27kip1 (CDKN1B).	Pancreatic islets from MEN1-deficient mice [216]
DAXX/ATRX/ ALT	DAXX ATRX	DAXX and TRX form a histone chaperone complex to deposit histone variant H3.3 at the telomeres and pericentric heterochromatin regions of the genome. They are frequently mutated in PNET samples.	Human PNET samples, Hela cells [217,218]
CDKN2A/RB1	RB1	Histone demethylase retinoblastoma binding protein 2 (Rbp2) was found overexpressed in PNETs. Aberrant expression of Rbp2 altered histone demethylation and contributed to PNET pathogenesis.	PNET patient samples, βlox5 cell, H727 cell, QGP-1 cell [192]
Notch	Notch1	HDAC inhibitor causes increased Notch 1 expression in tumor cells and mouse tumor xenografts [116,219]	BON-1 cells [219], carcinoid cancer cells, and mouse tumor xenografts [116]
SSTR2	SSTR2	Histone acetylation present on SSTR2. In addition, the combination treatment of HDACi (VPA) and camptothecin-somatostatin conjugate significantly reduced tumor growth compared to monotherapies.	BON-1 and QGP-1 cells [210,211], BON-1 xenograft mouse model [212]

**Table 3 ijms-25-01331-t003:** NcRNAs in PNETs. TKs include IGF, FGF, VEGF, EGF, HGF, and PDGF.

Signaling Pathway	Signaling Molecules	Non-Coding RNA Status	Experimental Systems
MEN1	MEN1	Menin negatively regulates miR-24-1 in a negative feedback loop manner.	BON-1 cells [232,233]
		MiR-24 negatively regulates menin in the endocrine pancreas.	MIN6 cells, βlox5 cells; floxed MEN1 mouse model [289]
		Menin upregulates the expression of MEG3.	Mouse insulinoma cells [231]
PTEN/PI3K/ AKT/mTOR/ c-Myc/TSC/ PRK ^1^	PTEN	MEG3 causes decreased p-PI3K, p-AKT, p-mTOR, and smaller tumor size.	Human retinoblastoma cells [290]
	PI3K	miR-144 causes decreased PTEN.	xenograft mouse model [238]
	AKT	MiR-144 correlated with increased p-AKT.	MIN6 cells [238]
	mTOR	IncRNA H19 causes increased PI3K-AKT and PNET progression.	Human insulinoma samples, QGP-1, PNET primary cells. QGP-1 xenograft model [239]
	HGF/MET	MEG3 downregulates c-MET in PNET.	MIN6 cells, mouse, and PNET patient samples [231].
HIF-1α/VHL	HIF-1α	MiR-210 expression is positively correlated with PNET progression and was shown to regulate colorectal adenocarcinoma progression through HIF1α.	PNET patient samples [244]
			FaDu head and neck cancer cell line, SU86.76 pancreatic cancer cell line, Xenograft mouse model [245,246].
RAS/MAPK/ NF1	RAS	MiR-431 promotes PNET progression by silencing DAB21P, resulting in the activation of the RAS pathway.	QGP-1 cell line, and xenograft mouse model [251].
ALT/DAXX/ ATRX	ATRX	ATRX negatively regulates miR-3653, which might serve as a risk factor for metastatic disease in PNETs.	Microarray differential expression of human PNET tissue samples [255].
Notch	Notch1,2,3, ASCL1	LncRNA XLOC_221242 is positively correlated with Notch/Wnt signaling.	PNET patient samples [272,291].
Wnt/β Catenin	Wnt, β-Catenin, SFRP1, WIF1	LncNEN885 negatively regulates Wnt/β-catenin signaling, leading to a reduction in EMT in PNETs. LncNEN885 is negatively correlated with PNET progression.	BON-1 cells, and PNET patient samples [276].
SSTR	SSTR2	The upregulation of miR-16-5p induces SSTR2 expression.	INS-1 cell line [284]

**Table 4 ijms-25-01331-t004:** FDA-approved anti-tumor epigenetic drugs and trials in PNET treatment. (DNMTi = DNA methyltransferase inhibitor; HDMi = Histone lysine demethylase inhibitor; HDACi = Histone deacetylase inhibitor; AML = acute myeloid leukemia; CML = chronic myelogenous leukemia; MDS = myelodysplastic syndromes; CTCL = cutaneous T-cell lymphoma; PTCL = peripheral T-cell lymphoma; MM = multiple myeloma; NR = no report; NA = not applicable).

Drug Name	Drug Target	Targeted Disease	Trial in PNETs	Status of Trial
Azacitidine	DNMTi	AML, CML, and MDS	NR	NA
5-Aza-2′-deoxycytidine	DNMTi	AML, CML, and MDS	Trial on synchronous AML and PNET	Treatment was successful with a combination of somatostatin analogs and decitabine, but with severe side effects [294].
Tazemetostat	HDMi	Advance epithelioid sarcoma	NR	NA
Enasidenib	HDMi	AML	NR	NA
Vorinostat	HDACi	CTCL	Pilot-imaging study to test the efficacy of vorinostat on radionuclide uptake.	A statistically significant increase in radionuclide uptake was observed [295].
Romidepsin	HDACi	CTCL and PTCL	Phase I trial of romidepsin in patients with pancreatic and other advanced solid tumors.	Stable disease status was observed when combined with treatment of gemcitabine [296].
Panobinostat	HDACi	MM	Phase II-trial against low-grade PNET	Fifteen patients were in the trial. No response was observed [297].
Belinostat	HDACi	PTCL	Phase I-trial against NET and small cell lung cancer	Partial response was observed when patients were treated with belinostat combined with cisplatin and etoposide [298].

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
