# Peer review of "Pancreatic Neuroendocrine Tumors: Signaling Pathways and Epigenetic Regulation"

_ijms, 2024, doi:10.3390/ijms25021331_

Round 1
Reviewer 1 Report
Comments and Suggestions for Authors
Jan 4, 2024
Dear Mr. Nattapon Kuntip
Editor-in-Chief
International Journal of Molecular Sciences
Dear Mr. Nattapon Kuntip
I have checked the manuscript (title: Pancreatic Neuroendocrine Tumors: Signaling Pathways and Epigenetic Regulation). This manuscript seems to be valuable for focusing on epigenetic alterations in pancreatic neuroendocrine tumors. I enumerate some comments as follows.
Major points
1. Overall, the contents of ‘3. Epigenetic regulation of PNET-related signaling pathways’ are redundant. In particular, I feel the description of epigenetic alterations in malignancies other than PNET is large in volume. If possible, please simplify the description.
Minor point
1. RAF, MEK, EAK are lacking in MAPK pathway in Figure 1. Please add them.
2. Please change the word ‘ATX’ to ‘ATRX’ in the line 66, page 2 in Introduction section.
Very sincerely yours,
Comments on the Quality of English Languagenone
Reviewer 2 Report
Comments and Suggestions for Authors
The manuscript entitled “Pancreatic Neuroendocrine Tumors: Signaling Pathways and Epigenetic Regulation” is a comprehensive review paper focusing on the epigenetic alterations including DNA methylation, histone modifications, non-coding RNAs in pancreatic neuroendocrine tumors. This paper adds to the literature and is expected to be of broad interest to basic and clinically orientated scientists. Overall, it is well-written, covering an understudied topic in depth and therefore deserves to be published in IJMS. Below are two minor issues to be addressed:
1) does this review paper refer only to neuroendocrine tumors or does it include neuroendocrine carcinomas too?
2) In the Introduction the authors could briefly refer to the new classification of pancreatic neuroendocrine tumors according to W.H.O. (5th edition, i.e. neuroendocrine tumors grade 1 – 2 – 3).
Round 2
Reviewer 1 Report
Comments and Suggestions for Authors
none